# Sub-Lexical Processing of Chinese–English Bilinguals: An ERP Analysis

**DOI:** 10.3390/brainsci14090923

**Published:** 2024-09-16

**Authors:** Yihan Chen, Eleonora Rossi

**Affiliations:** Department of Linguistics, College of Liberal Arts and Sciences, University of Florida, Gainesville, FL 32611, USA; yihan.chen@ufl.edu

**Keywords:** sub-lexical processing, bilingualism, P200, N400

## Abstract

Previous research has established that bilinguals automatically activate lexical items in both of their languages in a nonselectivemanner, even when processing linguistic information in the second language (L2) alone. However, whether this co-activation extends to the sub-lexical level remains debated. In this study, we investigate whether bilinguals access sub-lexical information while processing in their L2. Thirty-two Chinese–English bilinguals and thirty-one English monolinguals completed an EEG-based semantic relatedness task, during which they judged whether pairs of English words were related in meaning or not (±S). Unbeknownst to the participants, the form (±F) of the Chinese translations in half of the pairs shared a sub-lexical semantic radical. This leads to four conditions: +S+F, +S−F, −S+F, and −S−F. This design, along with the comparison to English monolinguals, allows us to examine if bilinguals’ native language is activated at the sub-lexical level when they are exposed only to L2. The results revealed that both groups showed sensitivity to semantic relatedness, as evidenced by a greater N400 for semantic unrelated pairs than related pairs, with monolinguals eliciting a more pronounced difference. Bilinguals, on the other hand, exhibited a greater P200 difference compared to monolinguals, indicating greater sensitivity to the hidden Chinese radical/form manipulation. These results suggest that highly proficient bilinguals automatically engage in lexical co-activation of their native language during L2 processing. Crucially, this co-activation extends to the sub-lexical semantic radical level.

## 1. Introduction

Language co-activation is a well-documented phenomenon in bilingual processing. Ample data suggest that bilinguals activate both of their languages in a non-selective manner even when processing in one language alone [1]. This rampant co-activation has been shown to prevent bilinguals’ ability to effectively inhibit one language while processing the other [2]. Notably, bilingual language co-activation occurs across all proficiency levels, including the early stages of language acquisition [3], and extends across various linguistic domains, from phonetic–phonological [4,5] and lexico-semantic [1,6] to syntactic–sentence processing [7,8].

Lexical-level co-activation, in particular, has been extensively studied using both behavioral [9,10] and neuroimaging methods [6,11,12], primarily in English and other alphabetic languages. For example, Hoshino and Thierry [6] demonstrated that semantic priming occurs not only when interlingual homographs (e.g., “pie,” which means “foot” in Spanish) are related to their English meaning (e.g., apple) but also to their Spanish meaning (e.g., toe). This suggests that lexical co-activation of interlingual homographs is automatic for Spanish–English bilinguals. Studies have also explored lexical co-activation in languages with different scripts [13,14,15,16,17,18,19]. Degani et al. [15], for instance, compared Arabic–Hebrew bilinguals and Hebrew monolinguals on a Hebrew semantic relatedness task. They found that cognate primes facilitated, and false-cognate primes interfered with, related Hebrew targets for bilinguals. This indicates a simultaneous activation of both languages, even in a single-language context.

Despite these findings, very few studies have examined bilingual language co-activation at the sub-lexical level (i.e., the activation of a meaningful sub-portion of a word/character). Unlike English, Chinese characters (e.g., 湖 /hú/ ‘lake’) consist of a semantic radical (氵‘water’) that indicates a related meaning and a phonetic element (胡 /hú/) that suggests pronunciation [20,21]. This unique feature of Chinese semantic radicals allows for the examination of meaning processing at the sub-lexical level. Therefore, the current study investigates language co-activation at the sub-lexical semantic radical level in Chinese–English bilinguals, exploring whether they activate sub-lexical information even when processing in their second language (English) alone. In addition, using electroencephalography (EEG), a technique that measures the brain’s electrical activity with high temporal resolution, this study ensures the tracking of real-time rapid processes involved in sub-lexical processing that might not be observable with behavioral measures alone. 

In what follows, we introduce relevant theoretical models of bilingual lexical processing, review empirical studies on the implicit activation of the first language (L1) during L2 processing, and outline how sub-lexical information is encoded in Chinese orthography. We will identify gaps and limitations along the way and then cast predictions on how the current design and data may inform us about bilingual sub-lexical processing.

### 1.1. Models of Bilinguals Lexical Processing

Several models have been proposed accounting for bilingual lexical co-activation, including the Bilingual Interactive Activation plus Model (BIA+) [22], the Revised Hierarchical Model (RHM) [23], and the more recent Multilink Model [24]. All of these models propose an integrated semantic/concept representation between languages and posit that this mental representation can be activated by both L1 and L2 word forms. BIA+ and Multilink further incorporate sub-lexical information and explain how they contribute to higher-level processing. However, these models are primarily based on alphabetic languages with the same writing scripts (i.e., Dutch and English) and shed little light on the underlying mechanisms for languages with different scripts. 

To address this limitation, Wen and van Heuven proposed a Chinese–English Interactive Activation (CE-IA) model, which assumes no inhibition between different script languages. Degani et al. [15] extended this cross-script model by examining sub-lexical co-activation in Arabic and Hebrew. Their model suggests that orthographic and phonological overlap varies across scripts. It also proposes that (sub-)lexical orthography and phonology can activate language membership nodes, but this activation is unidirectional, reflecting the persistence of cross-language influence in the bilingual lexicon regardless of script differences. However, this model does not specify whether the orthographic input from one language will trigger or suppress the orthographic activation of the other language at both lexical and sub-lexical levels. Our current study could offer further insights into these issues.

### 1.2. Implicit L1 Activation While Reading in a Second Language

Most relevant for this study is the empirical evidence for the implicit activation of the L1 when only the L2 is explicitly presented. Numerous previous studies have tested bilingual language co-activation using cross-language paradigms when both languages are overtly presented. These include designs using cognates [14,25,26], interlingual homographs [6,17], and translation priming [11,13,27]. Such paradigms, however, according to Thierry and Wu [1], create an artificial dual language environment in which both languages are inevitably activated. Instead, to argue for a ‘pure’ non-selective nature of the bilinguals’ mental lexicon, it is necessary for the L1 to be continuously activated even when all the experimental paradigms are presented exclusively in the L2.

An early attempt to measure bilingual lexical co-activation in this pure L2 environment is reported in Marian and Spivey [9]. They used eye-tracking to examine spoken language processing with Russian–English bilinguals. While listening to an audio sentence prompt, participants were presented with four picture items- including a target item (e.g., *shovel*), a within-language phonological distractor item (e.g., *shark*), a between-language phonological distractor (*sharik*, ‘balloon’) and an unrelated item. Results showed that Russian–English participants looked at both types of distractors more often than the control items and demonstrated that bilinguals constantly kept both languages activated during listening. Morford et al. [10] extended this paradigm to bimodal bilinguals. With an English (L2) semantic relatedness task, participants were asked to judge if two English words were related in meaning. Critically, half of the word pairs within each semantic group were selected such that they shared related forms when translated into American Sign Language (hidden factor). The results revealed that participants spent less time making the decision when the ASL translation of the English words shared related forms than the ones that did not share related forms. Both behavioral studies suggested that bilingual speakers accessed the L1 translation implicitly even if it was not required by the task. 

An EEG extension of these behavioral studies was reported by Thierry and Wu [1]. During the EEG task, Chinese–English bilinguals were presented solely with English (L2) words. Half of the word pairs were chosen such that they shared a character when translated into Chinese (i.e., train **火**车 -ham **火**腿), and the other half did not. Using this implicit paradigm, besides the semantically related effect elicited by a reduced N400 at a typical 350–600 ms time window, they were also able to reveal an effect of the hidden Chinese character repetition at an earlier time window (350–450 ms) that the authors interpret as an early N400. This character repetition effect did not interact with the semantic relatedness factor. These ERP results were in line with previous behavioral data showing that bilinguals indeed activated both languages even when they are processing in one language alone but also highlighted early neural signatures of these processes. Overall, however, the studies conducted so far have only focused on the whole word level, and they do not address whether and to what extent this co-activation extends to the sub-lexical level.

### 1.3. Sub-Lexical Encoding in Chinese Orthography

Chinese represents a great testbed to evaluate this hypothesis, as it offers the opportunity to examine sub-lexical activation. As mentioned before, Chinese characters are formed by joining together a semantic radical and a phonetic element. For instance, the word 湖 /hú/ ‘lake’ is composed of the semantic radical 氵that means ‘water’ and a phonetic element 胡 /hú/ that provides the sound. Critically, the semantic radical 氵‘water’ can also be found in other water-related words, such as 河 ‘river’, 海 ‘sea’, 汤 ‘soup’, 酒 ‘liquor’. According to Shen and Ke [28], semantic radicals are defined as ‘the smallest meaningful orthographic units that encode semantic information’. Three very important features of semantic radicals are that (1) they usually cannot stand alone to form a character, (2) they are recurring structures indicating categorical meaning, and (3) most of them do not have phonology associated with them. Shu and Anderson [29] estimated there are approximately 200 semantic radicals in Chinese and more than 80% of all modern Chinese characters consist of radicals. Testing Hong Kong elementary school students, Shu and Anderson [29] found that as children became more mature readers of Chinese, they began to develop an increasing awareness of this meaning-conveying function of semantic radicals. Students could both decompose a character into its radicals and use radicals to infer the meanings of unfamiliar characters. Su and Kim [30] further examined adult L2 learners of Chinese and showed that the productive knowledge of semantic radical function was positively related to word recognition. Moreover, Taft et al. [31] emphasized that semantic radicals are still an important processing unit for adult skilled readers. 

Although semantic radicals are a very prominent feature of the Chinese writing system, the meaning of the radical may not always match the meaning of the whole word. For example, 法 ‘law’ has nothing to do with water, but it still contains the radical 氵‘water’. This difference provides a good opportunity to study sub-lexical processing in written language.

Previous studies on Chinese word processing have examined this sub-lexical semantic radical level. Williams and Bever [32] investigated the facilitation effect of semantic radicals during word recognition. They found that words with semantic radicals matching their meanings led to faster reaction times, while mismatches resulted in slower responses, indicating potential conflict. In their second experiment [32], participants encountered characters with a blurred section. Errors and response times increased when the blur covered the semantic radicals compared to other parts of the word. Wang et al. [33] conducted an EEG-based lexical decision experiment with words whose semantic radicals were either related to the overall meanings or not. The ERP results revealed a facilitation effect, with a smaller P200 and larger N400 for words whose meanings matched their semantic radicals.

Importantly, all the studies mentioned so far have focused solely on the facilitation effect of semantic radicals in native monolingual Chinese speakers. Furthering this research, Chen and Perfetti [13] explored whether bilingual Chinese–English speakers are sensitive to sub-lexical semantic radicals in a mixed language context. They [13] employed both an implicit (color judgment) and an explicit reading task (word naming) to test whether a Chinese meaning equivalent character and its semantic radical were activated. Native Chinese speakers were instructed to read English primes (e.g., mouth ‘口’) and Chinese targets that fell into one of the four categories: (1) translation equivalent (e.g., 嘴 ‘mouth’; the semantic radical here is the 口 on the left side of the character, which means ‘month’), (2) both form- and semantically related (e.g., 唱歌 ‘sing’), (3) form-related but semantic unrelated (e.g., 喷泉 ‘fountain’), and (4) both form- and semantically unrelated (e.g., 铁路 ‘railway’). Results demonstrated that the Chinese meaning equivalent character is activated during L2 reading. However, this activation did not extend to the sub-character level. A potential explanation of this null result is that accuracy and reaction times captured at the end of each item may not grasp a real-time full picture of online language processing. Instead, neurophysiological methodologies such as EEG, with millisecond precision, will offer greater sensitivity and supplement behavioral results. This leads to our decision to integrate both behavioral and EEG measures in the current study. 

### 1.4. Current Study: Rationale and Predictions

The goal of the current study is to (1) determine whether Chinese–English bilinguals activate their native language (Chinese) even when their only input is L2 (English), and (2) evaluate the depth of this effect, specifically asking if the observed activation extends to the sub-lexical semantic radical level. To achieve these goals, Chinese–English bilinguals and English monolinguals completed a semantic relatedness task exclusively in English while their EEGs were recorded. Behaviorally, we predict that all participants will show better accuracy and shorter reaction times for semantically related pairs compared to unrelated ones, with monolinguals having a better performance than bilinguals. Additionally, since Chinese–English bilinguals are the only group that can be sensitive to the hidden Chinese form manipulation, we predict that they will have better accuracy and shorter reaction times for form-related word pairs compared to form unrelated word pairs.

Regarding the EEG signatures, two specific event-related potential (ERP) components are predicted to be observed in the context of the current paradigm, mainly the P200 and the N400. 

More specifically, the P200 is an early positive deflection that has been previously identified as an index for orthographic processing [34,35]. For example, Liu et al. [34] revealed that a reduced P200 was elicited when two words were only orthographically similar, but not when they were only phonologically or semantically similar. The authors interpreted this reduction in P200 as reflective of an inhibitory process, as characters sharing a radical but differing in pronunciation cause interference in generating pronunciation. In a later study, Zhang et al. [36] minimized the overt presentation of Chinese orthography and used a picture–character matching task. The results revealed a reduction in the P200 when the characters shared sub-lexical orthographic information with the paired pictures regardless of the word meaning (i.e., +O+S 
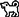
(狗)--狼 ‘dog-wolf’; +O−S 
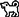
(狗)-猜 ‘dog-guess’). They argued that less processing effort is needed for orthographically similar characters as the corresponding picture name would co-activate characters sharing the same radicals, contributing to facilitation in orthographic processing. Although the exact underlying cognitive function remains unclear, a consistent finding revealed by these studies is that a reduced P200 is associated with orthographically similar words. Based on these previous results, for the current study, we predict a reduction in the P200 for Chinese–English bilinguals when processing English word pairs whose Chinese translation equivalents share a semantic radical, compared to those without a shared semantic radical.

The N400, on the other hand, has been intensively documented as a component that reflects the processing of semantic information. The N400 reflects the degree of semantic integration between a given word and its provided context. This context can be a single word [37], a sentence [38,39], a discourse [40], or even previous world knowledge [41]. Across these studies, a reduction in the N400 can be repeatedly observed when the target word is semantically, associatively, or categorically related to the previous prime(s) (see a review in [42,43]). Based on this extant literature, for the current paradigm, it is predicted that semantically related words will yield a reduced N400 compared to unrelated word pairs across both language groups. In addition, it is predicted that English monolinguals will exhibit a greater N400 difference than bilinguals, reflecting greater depth in semantic processing.

## 2. Methods

### 2.1. Participants

Seventy-four participants were recruited at the University of Florida. They included 40 Chinese–English bilinguals and 34 English monolinguals. All were between 18 and 35 years old (mean = 22.7 years; female (*n*) = 51, male (*n*) = 23), had normal or corrected-to-normal vision and hearing, and had no history of neurological disorders. The protocol was approved by the Institutional Review Board (IRB) of the University of Florida. Age, level of education, and handedness (right) were controlled across language groups. Chinese–English bilinguals were first exposed to English after the age of six. Although they are highly proficient in both languages, their proficiency remains unbalanced, with Chinese being stronger than English. Among the tested participants, data from 8 bilingual participants and 3 monolinguals were excluded from the final analysis due to low response accuracy or excessive EEG artifacts (detailed exclusion criteria are presented in Section 2.4. EEG Recording and Preprocessing). Table 1 summarizes the demographic and language background data from both groups in the final sample. The demographic information was obtained via a questionnaire completed in Qualtrics. The language background and self-reported proficiency data were collected using Language History Questionnaire 3 (LHQ 3, [44]). In addition, participants’ language proficiency was evaluated objectively using a verbal fluency task. 

### 2.2. Material and Design

#### 2.2.1. Semantic Relatedness Task

A semantic relatedness task was used in this study, where participants were asked to judge if two words were related in meaning. This task helps researchers understand how the brain processes semantic information and reveals how words/concepts are organized in the mental lexicon. A total of 192 English word pairs were generated for the semantic relatedness task. For each word pair, the first word consisted of the English translation of the meaning of the semantic radical, and the second word fell into one of the following relationships with the first word: semantically related and form-related to the semantic radical (+S+F), semantically unrelated and form-related to the semantic radical (−S+F), semantically related and form unrelated to the semantic radical (+S−F), and semantically unrelated and form unrelated to the semantic radical (−S−F) (examples shown in Table 2 below). All items were divided randomly into 4 counterbalanced lists with 48 word pairs each. Participants were randomly subdivided into one of the lists. A complete list of stimuli is presented in Appendix A. Altogether, a 2 × 2 × 2 design was used with semanticity (meaning related vs. meaning unrelated) and form (form related vs. form unrelated) as within-subject factors, and language group (bilinguals vs. monolinguals) as a between-subject factor. 

All experimental items were matched for word frequency (shown in Table 3) and semantic relatedness. Word frequencies of English were retrieved from the CLEARPOND database [45]; frequencies of Chinese translations were calculated using the Text of Recent CHinese (TorCH) 2009 and 2014 corpora [46]. There was no significant difference in word frequency of the target word among the four conditions in both English and their Chinese translations (both *p* > 0.1). English items across conditions were balanced for word length in both orthography and phonology (both *p* > 0.5), number of syllables (*p* > 0.5), and number of morphemes (both *p* > 0.1). All Chinese translations across conditions were balanced for their number of syllables/characters (*p* > 0.1)

To test the semantic relatedness of these word pairs, two separate semantic relatedness ratings were conducted: one with human raters and the other with a Natural Language Processing toolkit [41,42] (results presented in Figure 1). In the human rating test, 35 Chinese native speakers (mean age range: 18–29) completed a Qualtrics task in which they were presented with one pair of words each time and asked: “how much do you think these two words are related?”. They were instructed to answer on a Likert scale from 1 to 7 (with 1 being ‘not related at all’ and 7 being ‘highly related’). In the task, each word was compared to the meaning of the semantic radical rather than the form since a semantic radical rarely stands alone by itself as a character. For example, 海洋 ‘ocean’ was compared to the meaning 水 ‘water’ rather than the semantic radical 氵 ‘water’. Results demonstrated that there was a significant difference in semantic relatedness between +S and −S conditions regardless of their forms (or sub-lexical semantics) (all *p* < 0.001) and there was no significant difference within semantic conditions, that is, +S+F vs. +S−F and −S+F vs. −S−F (both *ps* > 0.1). 

The semantic relatedness was also calculated using NLP toolkits [47]. The semantic relatedness between English words was generated using Wup WordNet Similarity [48,49]. Under Wu and Palmer’s framework [50], word relatedness was calculated based on the depths of the two synsets in the WordNet taxonomies, along with the depth of the LCS (Least Common Subsumer). The semantic relatedness between Chinese words was collected using Chinese Open Wordnet. For both Chinese and English stimulus sets, the relatedness score was between 0 and 1 (with 0 being unrelated and 1 being highly related). Results were similar to those of human raters, as shown in Figure 1 (right). For both Chinese and English translations, a significant difference in semantic relatedness was found between +S and −S conditions regardless of their forms (all *p* < 0.001), and there was no significant difference between +F and −F within each semantic condition.

During the semantic relatedness task, participants were presented with stimuli using Eprime 3.0 Software [51] while their EEG was recorded. After a 500 ms fixation sign ‘+’ in the center of the screen, the first word was presented for 500 ms, followed by the second word after a variable interstimulus interval (ISI) of 300, 350, or 400 ms. The second word was also displayed for 500 ms, followed by another ISI of 300, 350 or 400 ms. Participants were then prompted with ‘???’ to answer whether the two words were related in meaning or not by pressing a response button on the Chronos box [51]. The motivation for using varying ISI is that it helps eliminate the temporal predictability of the item presentations and enables a greater attention span [52]. Participants were instructed very clearly to wait for the prompt first and then respond. Both accuracy and reaction time were collected. Figure 2 shows a sample trial of +S+F.

#### 2.2.2. Translation Task

After completing the semantic relatedness task, bilingual participants were asked to translate all the word stimuli they had just seen in the semantic relatedness task into Chinese. This task was completed to ensure that the translation participants reported was the same as the one we expected for our stimuli. During the translation task, all the items were presented in a random order one at a time. Participants were instructed to provide the Chinese translation verbally after each item. They were asked to speak loudly and clearly directly into the microphone [51] to ensure sound quality. If they did not know the Chinese translation, they were instructed to remain silent until the next word was presented. 

### 2.3. Study Procedure

To maintain consistency across participants, all surveys and tasks were administered in English only for all participants regardless of the language group (i.e., Chinese–English bilingual, English monolingual). After participants arrived at the lab, they first completed two online questionnaires: one gathered their basic information on health, handedness, and other demographics, and the other collected detailed language history background information using the LHQ3. Following the completion of the two questionnaires, participants moved into a soundproof, dimly lit experimental booth facing a computer monitor and were fitted with an electrode cap. Then, participants completed the main semantic relatedness (SR) task in English while their EEG was recorded. After the SR task, Chinese–English bilinguals completed a translation task in which they needed to translate all the words used in the SR task back to Chinese orally.

### 2.4. EEG Recording and Preprocessing

The final EEG analysis was conducted only on those participants with 70% and above behavioral accuracy in the semantic relatedness task for both groups. This results in the loss of 3 bilinguals and 3 monolinguals. Moreover, for the bilingual group only, 85% and above accuracy in the translation task was used as a cutoff for inclusion in the EEG data analysis, which led to the loss of an additional 4 participants. In total, 7 bilingual and 3 monolingual participants were eliminated from the final analysis. As such, the final sample that entered EEG data analysis is 33 bilinguals and 31 monolinguals. 

Continuous EEG data were acquired with an array of 32 Ag/AgCl scalp electrodes using BrainVision Products [53] active electrodes organized in accordance with the 10–20 system. Additionally, vertical and horizontal eye movements were measured using two sets of bipolar electro-oculograms (horizontal and vertical EOG). These electrodes were placed above and below the left eye (vertical) and on the right and left canthus (horizontal). An online reference electrode was placed on the right mastoid, and another was placed on the left mastoid for later re-referencing. Impedances were maintained <10 kΩ before and throughout the recordings. The signal was amplified using a Brain Vision actiCHamp amplifier with a 24-bit analog to digital conversion [53] was continuously recorded at a 1000 Hz sampling rate without online filters. All data were pre-processed offline using Brain Vision Analyzer 2 [54]. They were first manually checked to ensure general data quality. EEG data were re-referenced to the average of both mastoids and filtered using a 0.1–30 Hz IIR Butterworth filter with a 24 dB slope. After re-referencing, independent component analysis (ICA) was applied to identify and remove vertical and horizontal eye movements. After ICA, the data were subjected to a final inspection. All final artifact rejection was performed using a manual mode with visual confirmation. Participant data with artifact rejection rates greater than 20% were excluded from analysis, resulting in the loss of 1 bilingual participant (bringing the final analyzable sample for the bilingual group to n = 32). The overall rejection rate of the bilingual group is 2.9%, and that of the monolingual group is 0%.

### 2.5. Data Analyses

For the main semantic relatedness task, both behavioral and EEG measures were analyzed. Accuracy and reaction times were measured based on button press responses after the ‘???’ (green-button-related, red-button-unrelated) during EEG data collection. A generalized linear mixed effect model was used to analyze accuracy performance (all statistical analyses for the semantic relatedness task were performed using the lme4 package version 1.1.35.1 [55] with R version 4.1.3 [56]). The fixed effects entered in the model were semanticity (deviation coded: +S as 0.5 and −S as −0.5), form (deviation coded: +F as 0.5 and −F as −0.5), and language group (deviation coded: monolingual as 0.5 and bilingual as −0.5) and their interaction. We included by-subject and by-item intercepts. In addition, semanticity and form were entered as by-subject slopes and language group as a by-item slope. No singularity or collinearity (maximum variance inflation factor = 1.21) issues were detected. 

For reaction times analysis, only accurate responses were used; this resulted in the loss of 14.0% of all data. Furthermore, responses between 0 ms and 3000 ms post-stimulus onset were considered valid and then they were filtered with the range of 2.5 standard deviations above and below the mean. This resulted in an additional loss of 0.3% (total loss: 14.3%, including +S+F: 9.1%; −S+F: 25.8%; +S−F: 11.4%; −S−F: 11.0%). Mean RTs were log-transformed to avoid the influence of skewed distribution. A linear mixed effect model was used on mean RTs. The fixed effects were semanticity (deviation coded: +S as −0.5 and −S as 0.5), form (deviation coded: +F as −0.5 and −F as 0.5), and language group (deviation coded: monolingual as −0.5 and bilingual as 0.5) and their interaction. We included the bobyqa controller to reduce convergence errors. Our model also included by-subject intercept and semanticity, form as by-subject slopes. No singularity or collinearity (maximum variance inflation factor = 1.02) issues were detected. 

For the ERP analyses, ERPs were time-locked to the onset of the second word of each pair, which is the earliest point where ERP components related to semantic relatedness can emerge. Mean amplitude ERP data were extracted from 200 ms pre-stimulus to 800 ms post-stimulus to enable analysis of the two main ERP components of interest. Two different time windows were selected for analysis: the P200 window was selected from 150–250 ms [27,57] on the frontal-central electrode sites (F3, FC5, FC1, F4, FC6, FC2, Fz, Cz, C3, C4) and the N400 window was selected from 350 to 600 ms [1,43] on the central electrode sites (C3, Cz, C4, Pz, CP1, CP2, FC1, FC2, Fz). Two separate linear mixed effect models were performed for each time window. The dependent variable was the mean amplitudes at each time window. For both models, the fixed effects were semanticity (deviation coded: +S as 0.5 and −S as −0.5) and form (deviation coded: +F as −0.5 and −F as 0.5), language group (deviation coded: monolingual as 0.5 and bilingual as −0.5) and their interaction. The initial random effects were by-subject intercept, while semanticity and form were chosen as by-subject (individual) slopes. The initial model for both the P200 (150–250 ms) and N400 (350–600 ms) had a singularity issue. To try to resolve the singularity issue, two methods were utilized to reduce the random slope: (1) dropping semanticity as it had the lowest variance and (2) eliminating both form and semanticity to ensure the simplicity of the model. Then, we performed a model comparison using the ANOVA function to compare these two reduced models. The results showed that there was no significant difference between these two models (*p* > 0.1) for either time window. Model (2) was then chosen for its better fit indicated by a lower AIC (Akaike Information Criterion) and BIC (Bayesian Information Criterion). Therefore, the final model for both time windows included semanticity, form, language group, and their interaction as fixed effects and subject as a random effect (intercept).

## 3. Results

### 3.1. Behavioral Results

Accuracy results are reported in Table 4 and Figure 3. There were significant main effects of semanticity (+S: mean accuracy = 89.9%, SE = 3.7% vs. −S: mean accuracy = 82.1%, SE = 4.8%), form (+F: mean accuracy = 83.0%, SE = 4.7% vs. −F: mean accuracy = 89.1%, SE = 3.9%) and language group (monolingual: mean accuracy = 88.4%, SE = 4.0% vs. bilingual: mean accuracy = 83.7%, SE = 4.6%). Participants were more accurate for semantically related pairs than semantically unrelated pairs, and more accurate for form-unrelated pairs than form-related pairs. Overall, monolingual participants had a higher accuracy than bilingual participants. Additionally, a significant interaction was observed between form and semanticity. A post hoc Tukey test on this interaction revealed that the effect of semanticity (that is, the difference between semantically related and unrelated pairs) was only significant in form-related conditions (+S+F: mean accuracy = 91.0%, SE = 3.6% vs. −S+F: mean accuracy = 74.9%, SE = 5.5%; b = 1.74, SE = 0.44, *p* < 0.001 ***) but not form-unrelated conditions (+S−F: mean accuracy = 88.9%, SE = 4.0% vs. −S−F: mean accuracy = 89.3%, SE = 3.9%; b = 0.01, SE = 0.5, *p* > 0.05). Lastly, a significant three-way interaction was observed between form, semanticity and language group. This suggests that the relationship between form and semanticity interaction varies depending on the language group. 

For reaction times (shown in Table 5 and Figure 4), there was a significant main effect of semanticity (+S: mean = 437.9 ms, SE = 27.0 ms vs. −S: mean = 498.7 ms). Participants spent a longer time responding to semantically unrelated pairs compared with semantically related pairs. We also found a significant main effect of form (+F: mean = 481.9 ms, SE = 31.2 ms vs. −F: mean = 454.7 ms, SE = 29.1 ms), showing that participants had longer response times for form-related pairs compared with form-unrelated pairs. No interaction was found between semanticity, form and language group.

### 3.2. ERP Results

#### 3.2.1. P200

The P200 results revealed a significant main effect of form, as detailed in Table 6. Specifically, form-related word pairs elicited a smaller (less positive) P200 compared to form-unrelated pairs, regardless of semanticity or language group (see ERP and topographic plots in Figure 5 and Figure 6). Contrary to our predictions, this effect was observed across both bilingual and monolingual participants. We had anticipated a form by language interaction, where the form effect would be evident only among bilinguals.

In addition, the results revealed a significant semanticity by language group interaction. A post hoc Tukey test was performed and revealed that the effect of semanticity (difference of semantically related vs. unrelated) on P200 was greater in monolinguals (+S = 1.18 μV vs. −S = 0.51 μV; b = 0.67, SE = 0.26, *p* < 0.06) than bilinguals (+S = 1.54 μV vs. −S = 1.64 μV; b = −0.06, SE = 0.25, *p* > 0.5). More specifically, monolinguals showed a greater (more positive) P200 in response to semantically related word pairs than to semantically unrelated pairs. This interaction suggests that monolinguals, compared to bilinguals, show early indices of semantic processing that emerge before the typical time windows associated with it such as the N400. 

#### 3.2.2. N400

As we predicted, the results of N400 (shown in Table 7) revealed a main effect of semanticity, showing that semantically related pairs have a reduced N400 compared to semantically unrelated ones (ERP and topographic plots are presented in Figure 5 and Figure 6). Furthermore, a significant interaction was observed between semanticity and language group, revealing that the effect of semanticity was more significant for the monolingual group (post hoc Tukey test: +S = 0.37 μV vs. −S = −1.77 μV; b = 2.14, SE = 0.33, *p* < 0.001) than for the bilingual group (+S = −0.46 μV vs. −S = −1.12 μV; b = 0.65, SE = 0.33, *p* > 0.1). This demonstrates that monolinguals were also more sensitive to the difference between semantically related and unrelated pairs than bilinguals at a later window. 

## 4. Discussion

The goal of the current study was to investigate whether bilingual speakers co-activate their two languages even when processing input solely in their L2 (English), and to determine whether this co-activation extends to the sub-lexical semantic radical level. To achieve this, Chinese–English bilinguals and English monolinguals completed an EEG-based semantic relatedness task in English. Crucially, the bilingual participants were unaware of a hidden manipulation, where Chinese translations of the English items either contained the same semantic radical or did not. Behavioral and ERP results revealed that both groups were sensitive to semantic relatedness. However, only the ERP results indicated that bilinguals exhibited greater sensitivity to the hidden Chinese radical/form manipulation, as evidenced by a larger P200 difference between form-related and form-unrelated pairs compared to monolinguals. These findings extend Degani et al.’s theoretical development [15] of the well-known BIA+ model [22] and its extension [58], suggesting that bilingual lexical co-activation can indeed extend to the sub-lexical level even when the non-target language is in a different script. The following sections present detailed analyses of the behavioral and ERP results and discuss the broader implications of these findings for current models of bilingual language co-activation.

### 4.1. Discussion of Behavioral Results

Behaviorally, the results demonstrated lower accuracy and longer reaction times for semantically unrelated pairs compared to semantically related pairs. This effect aligns with a substantial body of previous literature and is consistent with Spreading Activation Theory (e.g., [59,60,61,62]). According to this theory, the activation of one concept spreads outward to related concepts, with stronger relatedness receiving more activation than weaker ones. Thus, the better accuracy and shorter reaction times observed for semantically related pairs can be explained by the facilitation of word retrieval through the spreading activation within the semantic network. 

Regarding form, participants exhibited longer reaction times and lower accuracy for form-related pairs compared to form-unrelated pairs. The underlying reasons for this effect are challenging to disentangle using solely behavioral data, given that behavioral data alone can provide information on the aggregate of the final cognitive processes. By incorporating real-time online processing data obtained through EEG, we can gain a more comprehensive understanding of this form effect. 

In terms of language groups, monolinguals demonstrated higher accuracy than bilinguals, regardless of semanticity and form. This result is predicted, as monolingual participants performed the task in their native language, English, whereas bilingual participants, particularly the ‘unbalanced bilinguals’ in this study, faced greater difficulty processing their second language. No significant difference in RTs was identified between language groups. This null result of RTs was expected because of the design of our study. In order to avoid the artifacts produced by response-related hand movement (which could be a confound captured by EEG measurement), participants were instructed to hold their response until 800ms after the stimuli, when ‘???’ appeared on the screen. This delay in response time could explain why the differences in RTs between language groups were not significant. 

### 4.2. Discussion of ERP Results

#### 4.2.1. P200 Findings and Interpretation

In the P200 window, the data revealed a main effect of form, showing that the P200 amplitude was smaller (i.e., less positive) for form-related words than for form-unrelated words regardless of the language group. Additionally, an interaction between semanticity and language group was observed, with the difference in P200 amplitude between semantically related and unrelated words being more pronounced in monolinguals than in bilinguals.

This interaction suggests that monolinguals are more sensitive to semantic relatedness than bilinguals, which corresponds to the previous literature by Landi and Perfetti [63]. They found that P200 is related to reading abilities, with a larger P200 observed for semantically related pairs compared to unrelated pairs, and this difference is larger for skilled readers. In our study, the monolingual group, considered more skilled readers as they are reading in their native language, exhibited an increased P200 for semantically related pairs. This indicates that the related meaning in the +S pairs provides a greater facilitation effect on target word processing, suggesting that monolinguals are already aware of the semantic relatedness at this stage.

However, the main effect of form observed across both language groups was not entirely in line with our initial predictions. According to our predictions, only Chinese–English bilinguals should have been sensitive to the hidden form manipulation, as they would be the only group to activate the Chinese translation equivalents. Since English monolingual speakers do not speak Chinese, they should not show any sensitivity to Chinese forms.

There are a number of possible alternative explanations for this effect. The first is that there could be some features of the English items themselves that may have created an unpredicted confound. For example, items in the +F conditions (i.e., +S+F and −S+F) could have been more similar in their English orthography compared to items in the −F conditions (+S−F and −S−F). However, we can rule out this possibility, because all stimuli were controlled for word frequency, orthographic and phonological length, number of syllables, and number of morphemes (see Table 3). More importantly, we also calculated the orthographic similarity of these English item pairs using the Levenshtein edit distance [64]. Levenshtein edit distance is a measure of the minimum number of single-character edits (insertions, deletions, or substitutions) required to change one word into another. This measure is often used to quantify the similarity between two words or strings. Results show that there is no difference between +F and −F conditions (both *ps* >0.05; see raw values of orthographic similarity in Appendix A). This suggests that the source of the observed main P200 effect was most likely not derived from English form similarity. 

Another possibility for the observed main modulation of the P200 for monolingual English speakers could be the difference between conditions in other lexical properties, such as concreteness. Due to the massive difficulty in creating the items and matching them for all the above-mentioned features, it was unrealistic to control for all the possible lexical properties. However, several previous studies have shown that the P200 [21,65] may function as an early signature for an initial distinction between concrete and abstract words, though a more robust distinction is found in later time windows such as the N400 [66,67], N700s [68], and the late positive component (LPC; [69,70]). Modulations of the P200 due to concreteness have been shown in recent studies. For example, Tsang and Zou [71] conducted an ERP study on Chinese word recognition with a go/no-go lexical decision task. A total of 1020 two-character words and 204 two-character pseudowords were used with one of the manipulations being concreteness. Their results demonstrated that the effect of concreteness started early and remained persistently from 0 to 600 ms after the onset of the stimuli. More relevant to our study, they found that higher concreteness led to a more positive component in the 200–300 ms time window. Jin et al. [72] took one step further and explored pseudowords when they are associated with real words with different emotional valence and concreteness. They presented participants with a series of pseudoword–word pairs after they learned the associations between meaningless pseudowords and positive/neural words. Their task was to decide if the word was positive or neutral; however, unbeknownst to them, the concreteness of these words was also manipulated. Using this paradigm, they found that pseudowords associated with concrete words elicited a larger, more positive P200 than those associated with abstract words. 

In line with these recent data, a re-analysis of the properties of our stimuli was performed. Word concreteness was calculated using the R package doc2concrete version 0.6.0, which provides a domain-specific pre-trained classifier for concreteness in advice and feedback data [73]. Results revealed that items that were part of the −F conditions (+S−F, −S−F) have higher concreteness than items that composed the +F conditions (+S+F, −S+F) (+F = 3.91 vs. −F = 4.42, chi-squared = 15.225, df = 1, *p* < 0.001; see raw values of concreteness at Appendix A). This confounding factor may explain why both monolingual and bilingual groups exhibited a larger (more positive) P200 amplitude in response to −F (concrete) pairs compared to +F (abstract) pairs. It also accounts for why behaviorally both groups demonstrated shorter reaction times and higher accuracy for −F(concrete) pairs relative to +F(abstract) pairs [74,75,76].

#### 4.2.2. P200 Revisit for Bilingual Sub-Lexical Processing

However, if concreteness was the only factor that influenced the P200 effect, then we would expect the difference between concrete (−F) and abstract (+F) items to be stronger for monolinguals as they are processing items in their native language while bilinguals are processing in their second language, English. Moving away from the P200 effect, but in line with this hypothesis, we indeed found that the monolingual group showed a greater N400 difference relative to bilinguals, signaling overall enhanced sensitivity to English. For the P200, however, we observed the opposite, i.e., the difference between -F (concrete) and +F (abstract) word pairs was larger for the bilingual group (−F−(+F) = 0.63 μV) than the monolingual group (−F−(+F) = 0.19 μV). In other words, bilinguals showed a much greater P200 difference than monolinguals. This suggests that concreteness cannot be the only factor behind the enhanced P200 difference for bilinguals. Instead, we propose that bilinguals must have another source that also contributes to the observed greater P200 difference. We posit that this influence would very likely be due to the hidden orthographic manipulation of the sub-lexical semantic radical in Chinese. Consistent with this observation, previous research has reported that the P200 can be an index for orthographic and phonological processing effects. For example, Liu and Perfetti [34] manipulated the orthographic, phonological and semantic similarities of Chinese character prime–target pairs. They found that target characters sharing similar radicals with the prime produced a smaller P200 relative to unrelated controls. Zhang et al. [36] used a picture–character matching task and found a smaller P200 when the characters shared sub-lexical orthographic information with the paired pictures regardless of the word meaning (i.e., +O+S 
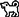
(狗)-狼 ‘dog-wolf’; +O−S 
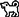
(狗)-猜 ‘dog-guess’). 

These studies combined illustrate that the P200 component can be used to track orthographic processing. The P200 is sensitive to both lexical and sub-lexical levels, exhibiting smaller amplitudes for orthographically similar words, regardless of differences in writing systems. While evidence consistently shows that the P200 captures orthographic similarities, the precise cognitive mechanism underlying this sensitivity remains a topic of debate. Some suggest that orthographically/phonologically similar words induce a facilitation effect [21,36,65] through spreading activation [60,77], whereby the processing of a word leads to activation spreading to related words in the mental lexicon. For example, the prime word 湖 ‘lake’, which is orthographically similar to the target word 河 ‘river’ as they share the same semantic radical 氵, facilitates the subsequent recognition of 河 ‘river’. Meanwhile, others may interpret it as a reflection of an inhibitory process necessary for managing the competition that arises when two words share the same orthography but differ in semantics. This interpretation is inferred from the study performed by Liu et al. [34], where such an inhibitory process was found when two words share the same orthography but differ in pronunciation, such as 凉 /liáng/ ‘cool;’ vs. 惊 /jīng/ ‘frighten’. This mismatch between orthography and pronunciation is the source of the interference when the participant has to generate a pronunciation, a process required for a pronunciation decision task.

Regarding our data, we propose that a less positive P200 in response to form-related pairs is a signature of an underlying faciliatory process that is at play when processing two words that share orthographic traits. More specifically, when the prime *water* was presented, it activated both the Chinese character 水 and the sub-lexical semantic radical 氵. Subsequently, 氵 initiated the spreading activation for orthographic neighbors, such as 湖 ‘lake’, 酒 ‘wine’, 海洋 ‘ocean’, and 法 ‘law’. By the time the target word was presented in English, if it was form-related (regardless of the meaning) to the prime when translated to Chinese; this provided early orthographic facilitation (captured by a smaller P200). We also want to emphasize that this orthographic facilitation effect is temporary (having disappeared by the time of N400). This transient effect was also reported by Perfetti and Tan [78] and Zeguers et al. [79], who observed an early facilitation effect of orthographic similarity, but it soon started declining [78,79] and became inhibitory [78]. We also found evidence of this later inhibitory effect from our behavioral results that bilingual participants spent longer time and were less accurate on +F pairs than -F pairs. Taken together, we believe that a smaller P200 reflects an underlying early faciliatory process when processing a target word that shares the same semantic radical with the prime. This shows that at an early stage (150–250 ms), bilinguals are already aware of the forms and meanings of the prime, their corresponding Chinese forms and their Chinese orthographic neighbors, and the form and possibly the meaning of the target in Chinese and English. Moreover, a faciliatory effect emerges when the pairs have identical forms (regardless of the meaning). However, they are not yet sensitive to the meaning relatedness.

#### 4.2.3. N400 and Semantic Processing

At the second time window, the results of the N400 revealed a main effect of semanticity, with a larger N400 found in semantically unrelated word pairs regardless of language group. This result corresponds to the large body of research showing that the N400 component is strongly indicative of semantic processing. Different experimental paradigms have been used in previous studies, such as words vs. pseudowords [80], cognates [81,82,83], homographs [6,17,84], and translation equivalents [11,13,27]. The languages tested included those that share the same script [25] or have different scripts [14] and those that are typologically close [25,82] and distant [13,27,85]. Across these studies, the results for semantic relatedness are unified: a larger, more negative N400 amplitude is elicited when the target is not semantically related to the prime, but a smaller N400 arises when the prime and the target are related. More importantly, our study shows that the effect of semanticity is much bigger and long-lasting for monolinguals than bilinguals. In previous research on bilinguals and the N400, different types of bilinguals were also represented including early [86] and late bilinguals [87,88], balanced [18] and unbalanced [89], proficient [18,82] and non-proficient L2 learners [90], and bimodal bilinguals [10], as well as using bilingualism from an individual difference perspective rather than using distinct categories [91]. Across these studies, the amplitude of the N400 is usually reduced for unbalanced bilinguals when they process their L2 relative to when they process their L1 [92].

### 4.3. Implications for Current Models of Bilingual Lexical Co-Activation

The results of the current study demonstrate that native-language activation is automatic for bilinguals, even when the L1 is not explicitly activated by the paradigm. More importantly, this activation can extend to the sub-lexical level. Figure 7 illustrates a cross-language lexical network of interactions within and across the two languages of both the prime and the target in the +S+F condition. In this figure, the black color represents English explicit activation, and the red color represents Chinese implicit activation. Within the prime, the English visual input ‘water’ activates its shared meaning, WATER, across the two languages because, according to the BIA+ model [22], bilinguals’ mental lexicon is mostly integrated. Consequently, the meaning WATER activates the Chinese orthography at both character/lexical level 水 ‘water’ and sub-lexical level 氵‘water’. Additionally, the English input ‘water’ may also directly activate the Chinese orthography. Through spread activation [59,60,61,62], this prime activates other words that are semantically related (the blue link) or form-related (the green link), providing an earlier start for the target words in the +S+F condition. Currently, both the semantic and form relationship between the prime and the target is marked by a solid line as they are related; these lines become dotted lines if they are unrelated.

The results of the current study may be relevant for bilingual processing models that take into account sub-lexical levels. Degani et al. [15] developed a model based on the famous BIA+ model [22] and BIA+ extension [58] and incorporated cross-script bilinguals. In this model, the sub(lexical) orthography and phonology are dynamic factors, with the flexibility of having a different degree of overlap ranging from no overlap (i.e., Chinese–English) to full overlap (i.e., Dutch–English). The semantic concept is mostly shared by both languages. Both the (sub)lexical orthography and phonology can activate language membership nodes. This activation is believed to be unidirectional, as argued by Degani and colleagues [15] using empirical results from Arabic–Hebrew bilinguals. The information on language membership accumulated in the language nodes is not sufficient to inhibit the activation of the non-target language. However, this model does not make explicit predictions for how the differences in scripts affect the orthographic activation at these two levels or whether the orthographic input of one language will activate/inhibit the orthography of the other language. Our current study gives more insight into these questions. The results suggested that the lexical level information was activated for the non-target language with a different script and that this activation can extend to the sub-lexical orthographic level, at least in the case of Chinese–English bilinguals.

Our findings also challenge the claims of the most recent Multilink model [24] for bilingual word recognition and translation. The Multilink model posits that there is no direct link between L2 and L1 word forms, suggesting instead that they are only connected through conceptual mediation. However, our results indicate that L1 form activation and shared concept activation may occur in parallel, particularly since L1 form activation was observed at an earlier time window (P200) even when reading L2 alone. This aligns with the Revised Hierarchical Model (RHM) [23] of bilingual lexical processing, which suggests a direct link between L2 and L1 word forms, but only for low-proficiency bilinguals. Our data further suggest that even highly proficient bilinguals maintain a direct link between L2 and L1 word forms.

## 5. Conclusions

This study provides direct observation that highly proficient bilinguals automatically engage in lexical co-activation of their native language even when processing their second language. More importantly, this co-activation can extend to the sub-lexical semantic radical level. These findings suggest that bilingual processing involves deeper, more automatic levels of language activation than previously understood, with implications for theories of bilingual cognition and language learning.

However, a limitation of this study is the relatively low number of word pairs per condition, which could affect the statistical power and robustness of the findings. Future research should address this by using larger corpora or datasets, such as the Tencent AI Lab Embedding Corpus, to enhance the generalizability of the results. Additionally, replication studies with traditional Chinese would be valuable to confirm the observed effects.

Building on these results, future research should also explore the effects of language immersion on co-activation by comparing bilinguals in different immersion contexts, such as Chinese–English bilinguals residing in China versus those in the United States. This could reveal the impact of environmental factors on native language activation. Another promising avenue for research involves investigating novel word learning and sub-lexical processing. An experimental design could involve creating new Chinese non-words paired with invented meanings, where half of the semantic radicals match the overall meaning, and the other half do not. Participants, including both Chinese native speakers and L2 Chinese learners, would participate in mini language learning sessions over several days. Testing their performance before and after these lessons would allow for the observation of early language acquisition and the role of semantic radicals in this process.

Overall, this research advances our understanding of the depth of bilingual language processing and sets the stage for further exploration of how language immersion and novel word learning impact language activation dynamics.

## Figures and Tables

**Figure 1 brainsci-14-00923-f001:**
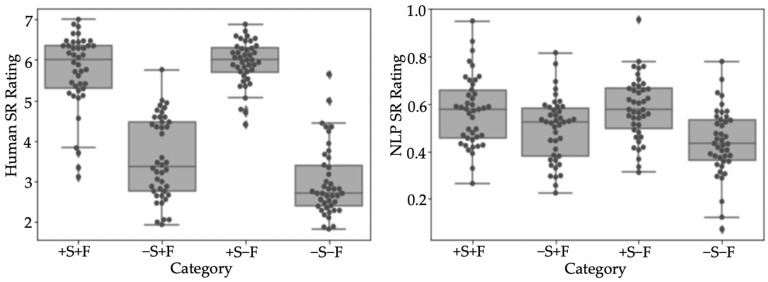
Semantic relatedness (SR) ratings for each condition. **Left**: box plot for human rating scores with a scale of 1(not related) to 7 (very related). **Right**: box plot for NLP rating scores with a scale of 0 (not related) to 1 (very related).

**Figure 2 brainsci-14-00923-f002:**
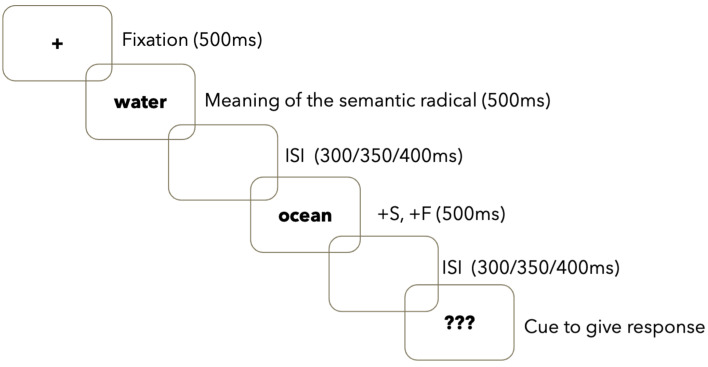
Sample Trial with an example of +S+F.

**Figure 3 brainsci-14-00923-f003:**
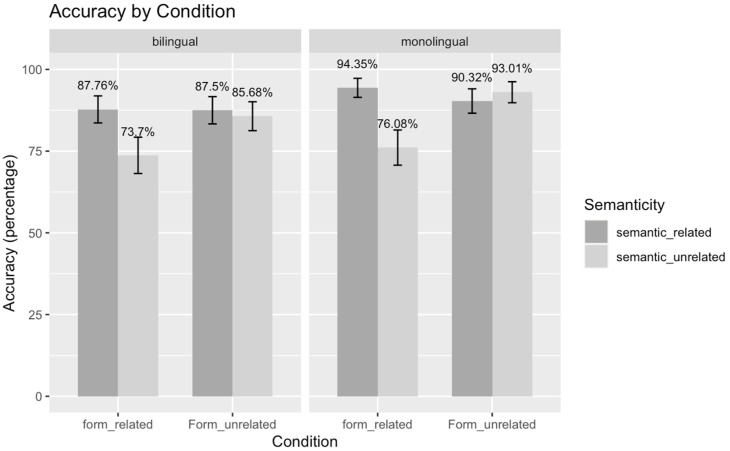
Behavioral results of accuracy for both language groups. Panels were used to indicate language group; x axis represents hidden form relatedness in Chinese and y axis shows accuracy in percentage. Colors were used to demonstrate semantic relatedness. Error bars depict standard errors in all cases.

**Figure 4 brainsci-14-00923-f004:**
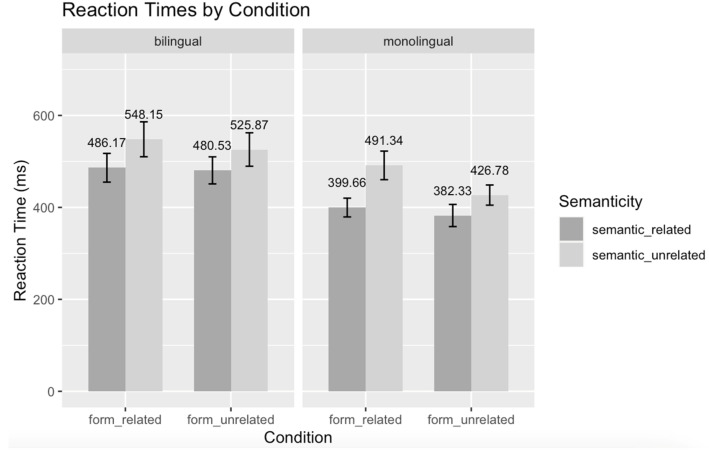
Behavioral results of mean RTs for both language groups (under revision). Panels were used to indicate language group; x axis represents hidden form relatedness in Chinese and y axis shows accuracy in percentage. Colors were used to demonstrate semantic relatedness. Error bars depict standard errors in all cases.

**Figure 5 brainsci-14-00923-f005:**
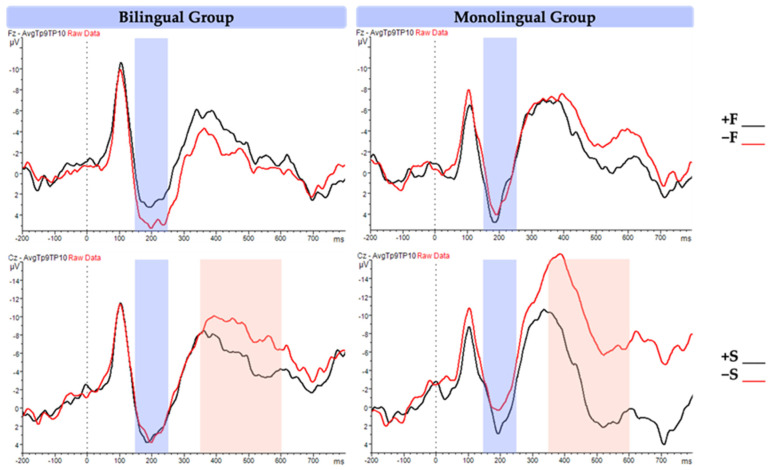
ERP results of the semantic relatedness task for both language groups. The x axis represents time in milliseconds and the y axis represents the mean amplitude in microvolts. The top two graphs represent the difference in form relatedness, and the bottom two graphs represent differences in semantic relatedness. The blue frame highlights the P200 time window (150–250 ms) and the red frame highlights the N400 time window (350–600 ms).

**Figure 6 brainsci-14-00923-f006:**
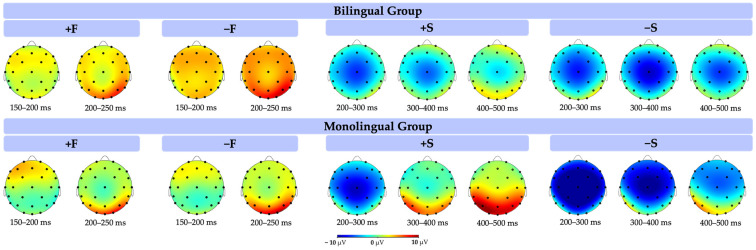
Topographic results of the semantic relatedness task for both language groups. +F: form related; −F: form unrelated; +S: semantic related; −S: semantic unrelated.

**Figure 7 brainsci-14-00923-f007:**
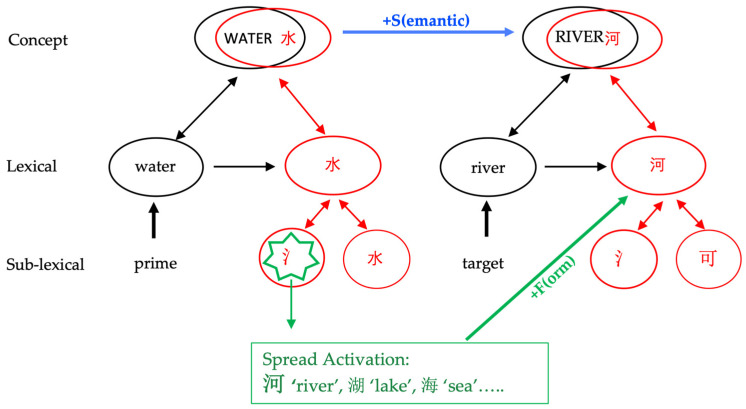
Illustration of English and (hidden) Chinese interaction when highly proficient Chinese–English bilinguals read English +S+F word pairs (adapted from [13]).

**Table 1 brainsci-14-00923-t001:** Characteristics of participants in the final analysis for both groups. The numbers in parentheses are the standard deviation of each group. Language proficiency is rated on a scale of 7.

	English Monolinguals (n = 31)	Chinese–English Bilingual (n = 32)	*p* Value
Gender	Male = 9	Male = 13	>0.1
Age (years)	19.2 (1.5)	26.1 (4.1)	<0.001 ***
Native language	English	Chinese	
English: age of acquisition (years)	2.0 (1.0)	8.4 (3.5)	<0.001 ***
Chinese: age of acquisition (years)	NA	2.0 (1.2)	
English proficiency: reading	6.8 (0.4)	5.2 (1.1)	<0.001 ***
English proficiency: listening	6.7 (0.5)	5.3 (1.2)	<0.001 ***
English proficiency: writing	6.7 (0.7)	5.1 (1.0)	<0.001 ***
English proficiency: speaking	6.7 (0.6)	5.1 (1.1)	<0.001 ***
Chinese proficiency: reading	NA	6.5 (0.9)	
Chinese proficiency: listening	NA	6.7 (0.5)	
Chinese proficiency: writing	NA	6.3 (1.2)	
Chinese proficiency: speaking	NA	6.6 (0.7)	
English daily usage (hours)	13.7 (5.0)	9.3 (5.3)	<0.01 **
Chinese daily usage(hours)	NA	6.0 (4.5)	
Verbal fluency (English) Word counts per 30 s	13.3 (3.6)	10.2 (1.6)	<0.001 ***
Verbal fluency (Chinese) Word counts per 30 s	NA	12.9 (2.2)	

Asterisks (**, and ***) are used to indicate the level of statistical significance of a *p*-value: *p* < 0.01 **, *p* < 0.001 ***.

**Table 2 brainsci-14-00923-t002:** Experimental design and stimuli conditions of the semantic relatedness task. All the word pairs were presented in English only (the Chinese translation is presented for reference). For each word pair, the first word represented the meaning of the semantic radical (‘water’ in this example) and the second word fell into one of the following four categories: +Semantic+Form, +Semantic−Form, −Semantic+Form and −Semantic−Form.

Semantic Radical: 氵Water Related	+Semantic	−Semantic
+Form	lake 湖	law 法
−Form	rain 雨	hand 手

**Table 3 brainsci-14-00923-t003:** Stimuli controlled for each condition. English words were controlled for word frequency, orthographic length, and phonological length. Chinese translations were controlled for word frequency and percentage of two-character words. Means and standard deviations (in parenthesis) are presented. *p* values are the results of one-way ANOVAs from comparing all four conditions.

	+S+F	−S+F	+S−F	−S−F	*p*-Value
Word Frequency/1M (English)	41 (68)	110 (203)	69 (174)	77 (88)	*p* > 0.1
Orthographic Length (English)	5 (2)	6 (2)	6 (2)	6 (2)	*p* > 0.5
Phonological Length (English)	4 (1)	5 (2)	5 (2)	4 (2)	*p* > 0.5
Num of Syllables (English)	1.7 (0.7)	1.8 (1.0)	1.8 (0.9)	1.5 (0.8)	*p >* 0.5
Num of Morphemes (English)	1.0 (0.2)	1.3 (0.6)	1.1 (0.3)	1.2 (0.4)	*p >* 0.1
Word Frequency/1M (Chinese Translation)	63 (104)	114 (211)	88 (106)	80 (93)	*p* > 0.1
Num of Syllables/Characters (Chinese Translation)	1.6 (0.5)	1.5 (0.5)	1.6 (0.5)	1.8 (0.4)	*p* > 0.1

**Table 4 brainsci-14-00923-t004:** Fixed effects of accuracy (note: × indicates an interaction relationship; The bolded ones are the factors that showed significant effect/interaction in the mixed effect model).

Accuracy	Estimate	SE	*p*-Value
Intercept	2.94	0.18	*p* < 0.001 ***
**S(emanticity)**	**0.88**	**0.35**	***p* < 0.05 ***
**F(orm)**	**−0.67**	**0.30**	***p* < 0.05 ***
**L(anguage Group)**	**1.04**	**0.28**	***p* < 0.001 *****
**S×F**	**1.72**	**0.57**	***p* < 0.01 ****
S×L	0.14	0.53	*p* > 0.05
F×L	−0.10	0.40	*p* > 0.05
**S×F×L**	**1.74**	**0.73**	***p* < 0.05 ***

Asterisks (*, **, and ***) are used to indicate the level of statistical significance of a *p*-value: *p* < 0.05 *, *p* < 0.01 **, *p* < 0.001 ***.

**Table 5 brainsci-14-00923-t005:** Fixed effects of mean RTs (note: **×** indicates an interaction relationship; The bolded ones are the factors that showed significant effect/interaction in the mixed effect model).

RTs	Estimate	SE	*p*-Value
Intercept	6.02	0.06	*p* < 0.001 ***
**S(emanticity)**	**−0.06**	**0.03**	***p* < 0.05 ***
**F(orm)**	**0.12**	**0.04**	***p* < 0.01 ****
L(anguage Group)	0.14	0.11	*p* > 0.05
S×F	−0.04	0.05	*p* > 0.05
S×L	0.07	0.06	*p* > 0.05
F×L	−0.09	0.08	*p* > 0.05
S×F×L	−0.01	0.10	*p* > 0.05

Asterisks (*, **, and ***) are used to indicate the level of statistical significance of a *p*-value: *p* < 0.05 *, *p* < 0.01 **, *p* < 0.001 ***.

**Table 6 brainsci-14-00923-t006:** Fixed effects of the P200 (note: **×** indicates an interaction relationship; The bolded ones are the factors that showed significant effect/interaction in the mixed effect model).

P200	Estimate	SE	*p*-Value
Intercept	1.21	0.22	*p* < 0.001 ***
S(emanticity)	0.28	0.18	*p* > 0.05
**F(orm)**	**0.41**	**0.18**	***p* < 0.05 ***
L(anguage Group)	−0.74	0.44	*p* > 0.05
S×F	0.28	0.37	*p* > 0.05
**S×L**	**0.77**	**0.37**	***p* < 0.05 ***
F×L	−0.44	0.37	*p* > 0.05

Asterisks (*, and ***) are used to indicate the level of statistical significance of a *p*-value: *p* < 0.05 *, *p* < 0.001 ***.

**Table 7 brainsci-14-00923-t007:** Fixed effects of the N400 (note: × indicates an interaction relationship).

N400	Estimate	SE	*p*-Value
Intercept	−0.75	0.23	*p* < 0.01 **
**S(emanticity)**	**1.40**	**0.23**	***p* < 0.001 *****
F(orm)	0.16	0.23	*p* > 0.05
L(anguage Group)	0.09	0.46	*p* > 0.05
S×F	0.87	0.47	*p* > 0.05
**S×L**	**1.49**	**0.47**	***p* < 0.01 ****
F×L	−0.40	0.47	*p* > 0.05
S×F×L	0.84	0.93	*p* > 0.05

Asterisks (**, and ***) are used to indicate the level of statistical significance of a *p*-value: *p* < 0.01 **, *p* < 0.001 ***.

## Data Availability

The data presented in this study are available on request from the corresponding authors due to privacy reasons.

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
