# Peer review of "Sub-Lexical Processing of Chinese–English Bilinguals: An ERP Analysis"

_brainsci, 2024, doi:10.3390/brainsci14090923_

Round 1

Reviewer 1 Report

Comments and Suggestions for Authors

Reviewer

Comments to the author:

Manuscript ID: brainsci-3170816

Title:Neurophysiological signatures of sub-lexical processing in Chinese-English bilinguals”

Thank you for the opportunity to review this manuscript. The issue of the manuscript is interesting and has significant value for researchers and clinicians who work in the field of bilingualism / multilingualism. 

Abstract

The abstract is unstructured, but according to the guidelines of the journal should be structured, containing the following headings: Background/Objectives, Methods, Results, Conclusions. Please reform properly.

Introduction

-After the description of the characteristics of the Chinese language, citations are needed.

-Please delete the goal of the study from page 2. You refer to your aim and the predictions you made in paragraph 1.2.

-Please explain what is L2 before you present the abbreviation.

-Is Marian and Spivey (2003). Please correct in your text.

-Please explain the abbreviation ASL.

-Page 4. In brackets citations with two authors are formed as following: Williams & Bever, 2010. Please correct the form of the citation.

-The authors must explain in their manuscript what is the meaning of the following abbreviations +O+S2, +O-S, -O-S, before they use them.

-Section 1.2 Please correct in the next phrase: “….for semantic related pairs compared wot unrelated ones…” the word “not”.

-Please, make sure that you consistently form the citations according to APA and Journal’s guidelines.

Methods

-In the participants section, you should describe if the two groups are similar in terms of age, gender, and level of proficiency. I strongly recommend adding a third column in Table 1, presenting the p values.

Results

The results are well-presented.

Discussion

Please delete the following phrase in the parenthesis (detailed discussion is presented below under Discussion: P200). It is unnecessary.

Who are called unbalanced bilinguals? Why do the authors characterize their bilingual sample that way? Does it have to do with the level of proficiency of each mother language? This kind of information belongs to the Participants section and not in the Discussion.

Overall, a well-written discussion.

Comments on the Quality of English Language

Some minor errors in English

Author Response

Thank you for the thoughtful and useful suggestions and comments. In what follows, we have addressed to the best of our abilities all the comments and suggestions. As a result, we feel that the manuscript has improved significantly.

Comment 1: Abstract-- The abstract is unstructured, but according to the guidelines of the journal should be structured, containing the following headings: Background/Objectives, Methods, Results, Conclusions. Please reform properly.

Answer 1:

Thank you for your suggestions. We revised our abstract accordingly as the following:

Background: Previous research has established that bilinguals automatically activate lexical items in both of their languages in a nonselective manner, even when processing linguistic information in the second language (L2) alone. However, whether this co-activation extends to the sub-lexical level remains debated. In this study, we investigate whether and to what extent bilinguals access sub-lexical information even when they are processing in their L2. Methods: Thirty-two Chinese-English bilinguals and thirty-one English monolinguals completed an EEG-based semantic relatedness task, during which they judged whether pairs of English words were related in meaning or not (±S). Unbeknownst to the participants, the form (±F) of the Chinese translations in half of the pairs shared a sub-lexical semantic radical. This leads to four conditions: +S+F, +S-F, -S+F, and -S-F. With this design and by comparing to English monolinguals, it allows us to examine if bilinguals’ native language is activated to the sub-lexical level when only exposed to L2. Results: A greater N400 for semantic unrelated pairs revealed that both groups were sensitive to semantic relatedness, with monolinguals eliciting a more pronounced difference. Bilinguals, on the other hand, elicited a greater P200 compared to monolinguals, indicating greater sensitivity to the hidden Chinese radical/form manipulation. Conclusions: These results reveal that highly proficient bilinguals automatically engage in lexical co-activation of their native language even when processing in their L2. Crucially, this co-activation extends to the sub-lexical semantic radical level.”

Comment 2: Introduction--After the description of the characteristics of the Chinese language, citations are needed.

Answer 2: Thank you for your feedback. We added the following two references on Chinese orthography:

[20]    Myers, J. The Grammar of Chinese Characters | Productive Knowledge of Formal Patterns in an Orthographic System, First Edition.; Routledge: London, 2019.

[21]    Hsu, C. H.; Wu, Y. N.; Lee, C. Y. Effects of Phonological Consistency and Semantic Radical Combinability on N170 and P200 in the Reading of Chinese Phonograms. Front. Psychol., 2021, 12. https://doi.org/10.3389/fpsyg.2021.603878.

Comment 3: Introduction--Please delete the goal of the study from page 2. You refer to your aim and the predictions you made in paragraph 1.2.

Answer 3: Thank you for your feedback. After careful consideration, we have decided to keep it as it is, and also to better address the feedback from Reviewer 2 and 3.

We believe it is an effective writing strategy to start with a general overview (Section 1.), then gradually break it down into more specific details (Section 1.1, 1.2 and 1.3), and finally summarize the key research question at the end (Section 1.4). In order to do this, we reorganized our Introduction as the following:

  1. Introduction (a brief introduction)
    • 1.1 Models of Bilingual Lexical Processing (summarized the models and their development and listed the limitations of the current models)
    • 1.2 Implicit L1 Activation while Reading in a Second Language (summarized the empirical findings and identified the gap)
    • 1.3 Sub-lexical encoding in Chinese orthography (stated why Chinese is a good option here for addressing the previous model limitation and empirical gap)
    • 1.4 Current study: Rationale and predictions

This approach not only provides a clear and logical flow to the writing but also helps to reinforce the main ideas for the reader, ensuring a thorough understanding of the topic, and it helps address the comments of the other reviewers regarding the Introduction.

Comment 4: Introduction--Please explain what is L2 before you present the abbreviation.

Answer 4:Thank you for pointing it out. We adjusted it accordingly (Page 2).

Comment 5: Introduction--Is Marian and Spivey (2003). Please correct in your text.

Answer 5:Thank you for pointing it out. We adjusted it accordingly (Page 2).

Comment 6: Introduction--Please explain the abbreviation ASL.

Answer 6:Thank you for pointing it out. We adjusted it accordingly (Page 2).

Comment 7: Introduction--Page 4. In brackets citations with two authors are formed as following: Williams & Bever, 2010. Please correct the form of the citation.

Answer 7: Thank you for spotting it. The citation style has been changed to numbers according to the journal’s requirements. So this is not applicable anymore.

Comment 8: Introduction--The authors must explain in their manuscript what is the meaning of the following abbreviations +O+S2, +O-S, -O-S, before they use them.

Answer 8: Thank you for your suggestion. We changed it into full-text description rather than abbreviations (Page 4).

Comment 9: Introduction--Section 1.2 Please correct in the next phrase: “….for semantic related pairs compared wot unrelated ones…” the word “not”.

Answer 9: Thank you for your suggestion, the sentence has been corrected to the following:

“…all participants will show better accuracy and shorter reaction times for semantic related pairs compared with unrelated ones.

Comment 10: Introduction--Please, make sure that you consistently form the citations according to APA and Journal’s guidelines.

Answer 10: Thank you. We have changed all the citations into numbers according to Journal’s guidelines.

Comment 11: Methods--In the participants section, you should describe if the two groups are similar in terms of age, gender, and level of proficiency. I strongly recommend adding a third column in Table 1, presenting the p values.

Answer 11: Thank you for your feedback. In response, we have added a third column to Table 1 to include p-values. Regarding gender, there is no significant difference between the groups. Although there is a significant age difference between them (monolingual = 19.2; bilingual = 26.1), both groups have reached similarly high levels of cognitive capacity. This has been supported by Icenogle et al. (2019), who measured both working memory and language performance (using a verbal fluency task) as aggregate indicators of cognitive capacity. A graph from their study is provided in the word document, along with the reference.

Icenogle G, Steinberg L, Duell N, Chein J, Chang L, Chaudhary N, Di Giunta L, Dodge KA, Fanti KA, Lansford JE, Oburu P, Pastorelli C, Skinner AT, Sorbring E, Tapanya S, Uribe Tirado LM, Alampay LP, Al-Hassan SM, Takash HMS, Bacchini D. Adolescents' cognitive capacity reaches adult levels prior to their psychosocial maturity: Evidence for a "maturity gap" in a multinational, cross-sectional sample. Law Hum Behav. 2019 Feb;43(1):69-85. doi: 10.1037/lhb0000315. PMID: 30762417; PMCID: PMC6551607.

Comment 12: Results --The results are well-presented.

Answer 12: Thank you!

Comment 13: Discussion--Please delete the following phrase in the parenthesis (detailed discussion is presented below under Discussion: P200). It is unnecessary.

Answer 13: Thank you for your comment. The corresponding text has been deleted.

Comment 14: Discussion--Who are called unbalanced bilinguals? Why do the authors characterize their bilingual sample that way? Does it have to do with the level of proficiency of each mother language? This kind of information belongs to the Participants section and not in the Discussion.

Answer 14: Thank you for your suggestions. We added the following text in the Participants section about ‘unbalanced bilinguals’ (Page 5).

Although they are highly proficient in both languages, their proficiency remains unbalanced, with Chinese being stronger than English.

 The bilinguals are categorized as unbalanced bilinguals because of their age of acquisition (AoA) and their proficiency.  In our study, the bilinguals were exposed to Chinese at the age of 0-2 and to English after the age of 6. In addition, according to the self-reported proficiency data, they believed themselves to be more proficient in Chinese (6.3-6.7 out of 7) than in English (5.1-5.3 out of 7). Therefore, although our bilinguals are highly proficient in both languages, their proficiency remains unbalanced, with Chinese being stronger than English.

Comment 15: Discussion--Overall, a well-written discussion.

Answer 15: Thank you so much!

Reviewer 2 Report

Comments and Suggestions for Authors
  • The abstract mentions the results in terms of ERP components (N400 and P200), but it doesn't explicitly state the broader implications or conclusions drawn from these findings.

    While the abstract does mention that the study investigates whether and to what extent Chinese-English bilinguals access sub-lexical information, it could be more explicit in stating the specific research question or hypothesis.

    The abstract describes the task (semantic relatedness task) but doesn’t clearly state how this task is related to the investigation of sub-lexical access.

    There is no mention of why this research is important or what gap it fills in the existing literature.

    Mentioning limitations or future research directions could enhance the abstract's comprehensiveness.

  • To improve your introduction, focus on streamlining the content for clarity and conciseness. Begin by clearly stating the phenomenon of language co-activation in bilinguals, then succinctly summarize key research findings and gaps, especially the focus on lexical-level studies in alphabetic languages. Emphasize the unique contribution of your study by highlighting its novel focus on sub-lexical processing in a non-alphabetic language like Chinese, and how this will extend current models of bilingual processing. Ensure that the logical flow connects these points effectively, leading to the rationale and objectives of your study.
  • The Methods section is generally well-organized and detailed, but there are a few areas where improvements could enhance clarity and readability. Firstly, the Participants section could benefit from a clearer explanation of the rationale behind the exclusion criteria and the final sample size. This would help the reader understand the decision-making process and the impact of these exclusions on the study's validity. Secondly, in the Materials and Design section, it might be helpful to provide a brief explanation of the significance of the Semantic Relatedness Task and why it was chosen, to give context to the reader who may not be familiar with this type of task. Additionally, the detailed description of the word pairs and their categorization could be made more concise or moved to supplementary materials, allowing the main text to focus on the core experimental design. In the Study Procedure section, consider elaborating on the rationale for using a dimly lit experimental booth, as this could influence participants' performance and should be justified. For the EEG Recording and Preprocessing section, it would be beneficial to explain the choice of a 0.1–30 Hz IIR Butterworth filter and how this range was determined to be optimal for the study. Finally, in the Data Analyses section, the reasoning behind the specific selection of time windows for ERP analysis could be more thoroughly discussed to strengthen the connection between the methodological choices and the study's hypotheses. Overall, these adjustments would make the methods more transparent and easier to follow for readers.
  • To enhance the Results section, consider a clearer and more structured presentation of findings by clearly separating and summarizing the key results of each type of data (behavioral, EEG) and their interactions. Begin with a concise summary of the main effects and interactions before diving into specific details. Highlight significant findings and their implications, using consistent terminology and straightforward language. Also, ensure that each result is directly linked to its corresponding table or figure, and interpret interactions in a way that clarifies their significance. For the EEG results, explicitly connect the findings to the hypotheses or research questions posed. This approach will make the results more accessible and easier to interpret for readers.

    To strengthen your discussion, focus on synthesizing your findings with existing literature and models more cohesively. Start by clearly stating how your results support or challenge previous theories of bilingual lexical processing, such as Spreading Activation Theory, the BIA+, or the Revised Hierarchical Model. Highlight the novel contributions of your study, particularly how the observed P200 and N400 effects in bilinguals provide new insights into the co-activation of lexical and sub-lexical representations across languages. Discuss the implications of these findings for understanding bilingual processing, emphasizing how your study extends current models by showing that sub-lexical orthographic activation can occur even when the L1 is not explicitly required. Additionally, address any limitations and propose specific directions for future research to further explore these mechanisms and their implications for bilingual language processing.

  • To strengthen and refine your conclusion, consider the following recommendations: Reiterate the main findings of your study in a concise manner. Ensure that the summary reflects the significance and implications of your results. For example: "This study reveals that highly proficient bilinguals engage in automatic lexical co-activation of their native language even when processing their second language, extending to the sub-lexical semantic radical level." Highlight the broader impact of your findings. Explain why this automatic co-activation is important and how it contributes to our understanding of bilingual processing. For example: "These findings suggest that bilingual processing involves deeper and more automatic levels of language activation than previously understood, with potential implications for theories of bilingual cognition and language learning."  Make your recommendations for future research specific and actionable. State clearly what aspects should be investigated and why. For instance: "Future research should explore the effects of language immersion on co-activation by comparing bilinguals in different immersion contexts, which could reveal the impact of environmental factors on native language activation." Connect the suggested new research directions to your study’s findings. For example: "Building on these results, investigating novel word learning and sub-lexical processing could provide insights into the mechanisms of early language acquisition and the role of semantic radicals in this process." End with a concluding statement that reinforces the importance of your study and its contributions. For example: "Overall, this research advances our understanding of bilingual language processing and sets the stage for further exploration of how language immersion and novel word learning impact language activation dynamics."

Comments on the Quality of English Language
  • The abstract mentions the results in terms of ERP components (N400 and P200), but it doesn't explicitly state the broader implications or conclusions drawn from these findings.

    While the abstract does mention that the study investigates whether and to what extent Chinese-English bilinguals access sub-lexical information, it could be more explicit in stating the specific research question or hypothesis.

    The abstract describes the task (semantic relatedness task) but doesn’t clearly state how this task is related to the investigation of sub-lexical access.

    There is no mention of why this research is important or what gap it fills in the existing literature.

    Mentioning limitations or future research directions could enhance the abstract's comprehensiveness.

  • To improve your introduction, focus on streamlining the content for clarity and conciseness. Begin by clearly stating the phenomenon of language co-activation in bilinguals, then succinctly summarize key research findings and gaps, especially the focus on lexical-level studies in alphabetic languages. Emphasize the unique contribution of your study by highlighting its novel focus on sub-lexical processing in a non-alphabetic language like Chinese, and how this will extend current models of bilingual processing. Ensure that the logical flow connects these points effectively, leading to the rationale and objectives of your study.
  • The Methods section is generally well-organized and detailed, but there are a few areas where improvements could enhance clarity and readability. Firstly, the Participants section could benefit from a clearer explanation of the rationale behind the exclusion criteria and the final sample size. This would help the reader understand the decision-making process and the impact of these exclusions on the study's validity. Secondly, in the Materials and Design section, it might be helpful to provide a brief explanation of the significance of the Semantic Relatedness Task and why it was chosen, to give context to the reader who may not be familiar with this type of task. Additionally, the detailed description of the word pairs and their categorization could be made more concise or moved to supplementary materials, allowing the main text to focus on the core experimental design. In the Study Procedure section, consider elaborating on the rationale for using a dimly lit experimental booth, as this could influence participants' performance and should be justified. For the EEG Recording and Preprocessing section, it would be beneficial to explain the choice of a 0.1–30 Hz IIR Butterworth filter and how this range was determined to be optimal for the study. Finally, in the Data Analyses section, the reasoning behind the specific selection of time windows for ERP analysis could be more thoroughly discussed to strengthen the connection between the methodological choices and the study's hypotheses. Overall, these adjustments would make the methods more transparent and easier to follow for readers.
  • To enhance the Results section, consider a clearer and more structured presentation of findings by clearly separating and summarizing the key results of each type of data (behavioral, EEG) and their interactions. Begin with a concise summary of the main effects and interactions before diving into specific details. Highlight significant findings and their implications, using consistent terminology and straightforward language. Also, ensure that each result is directly linked to its corresponding table or figure, and interpret interactions in a way that clarifies their significance. For the EEG results, explicitly connect the findings to the hypotheses or research questions posed. This approach will make the results more accessible and easier to interpret for readers.

    To strengthen your discussion, focus on synthesizing your findings with existing literature and models more cohesively. Start by clearly stating how your results support or challenge previous theories of bilingual lexical processing, such as Spreading Activation Theory, the BIA+, or the Revised Hierarchical Model. Highlight the novel contributions of your study, particularly how the observed P200 and N400 effects in bilinguals provide new insights into the co-activation of lexical and sub-lexical representations across languages. Discuss the implications of these findings for understanding bilingual processing, emphasizing how your study extends current models by showing that sub-lexical orthographic activation can occur even when the L1 is not explicitly required. Additionally, address any limitations and propose specific directions for future research to further explore these mechanisms and their implications for bilingual language processing.

  • To strengthen and refine your conclusion, consider the following recommendations: Reiterate the main findings of your study in a concise manner. Ensure that the summary reflects the significance and implications of your results. For example: "This study reveals that highly proficient bilinguals engage in automatic lexical co-activation of their native language even when processing their second language, extending to the sub-lexical semantic radical level." Highlight the broader impact of your findings. Explain why this automatic co-activation is important and how it contributes to our understanding of bilingual processing. For example: "These findings suggest that bilingual processing involves deeper and more automatic levels of language activation than previously understood, with potential implications for theories of bilingual cognition and language learning."  Make your recommendations for future research specific and actionable. State clearly what aspects should be investigated and why. For instance: "Future research should explore the effects of language immersion on co-activation by comparing bilinguals in different immersion contexts, which could reveal the impact of environmental factors on native language activation." Connect the suggested new research directions to your study’s findings. For example: "Building on these results, investigating novel word learning and sub-lexical processing could provide insights into the mechanisms of early language acquisition and the role of semantic radicals in this process." End with a concluding statement that reinforces the importance of your study and its contributions. For example: "Overall, this research advances our understanding of bilingual language processing and sets the stage for further exploration of how language immersion and novel word learning impact language activation dynamics."

Author Response

Please see the attached document below. 

Reviewer 3 Report

Comments and Suggestions for Authors

I would like to refer to the article entitled

Sub-lexical Processing of Chinese-English Bilinguals: An ERP Analysis

First of all, let me mention that you have done a very good and methodical preparation of the text, covering most of the theoretical background required. However, some points need reinforcement, and better organization so that the presentation is more documented, scaled, and clear.

Below I have notes for you to improve the text:

1. Correct the bibliography citation for formatting throughout the text following the journal's standard as it is now in APA and check that the bibliographic references are correctly cited.

2. The bibliography should also be modified in terms of formatting instead of alphabetical order, use the journal template.

3. I think you should strengthen the text in the points of the second paragraph of the introduction regarding the Chinese semantic element or radical and the EEG.

4. It is important to gather the aim and sub-objectives of your study at the end of the introduction before the methodology and beforehand to point out the gap in existing knowledge as you have already pointed out. Points of the purpose and objective of your study are scattered throughout the text of the introduction and make it difficult for the reader to clearly understand what will be studied and why.

5. In your methodology note the process of consent and recruitment of the population. If the population was assessed with some instrument regarding language knowledge in the respective language it should be noted as self-reporting through a questionnaire only weakens the initial assessment of the knowledge of the sample.

6. In 2.2.1 note by which criterion the language material was selected and whether you have relied on any previous study or protocol regarding the assessment with this material. In addition to the semantic content and frequency, it should be noted the complexity of the linguistic material in terms of the number of syllables and morphemes. It is important to support at the beginning of the paragraph if the same criterion (CLEARPOND and TorCH) also applies to the first condition of the English words, or if it does not apply, to cite similar documentation.

7. Note the literature on which the process of semantic relatedness was based and whether a similar process has been followed in another study based on the Natural Language Processing Toolkit

8. To define the NLP toolkit and note the bibliography accordingly as well as Eprime 3.0 Software.

9. Very good and structured presentation of the results, I suggest you follow a similar structure in the discussion of the results as the discussion needs restructuring. You have noted in some places that a detailed discussion will follow, for example about P200 but there is no corresponding sub-section in the text, or you mention section 5.4 in which N400 will be discussed but such a section does not exist. I suggest following the results structure "Behavioral Assessment – ​​Electrophysiological Assessment" and have there subdomains of P200, N400, and overall results. You must delete the points where you note that a detailed analysis will be done later in the text.

10. Important to note a separate subfield in the discussion of sublexical semantics

11. The fifth paragraph of the discussion needs to be bibliographically supported and strengthened with a newer reference and more details.

12. The last paragraph of the discussion needs support with as much contemporary literature as possible.

13. Before your conclusions, note the limitations of the study.

14. In your conclusions, answer all the questions raised in the purpose and individual objectives of the study, especially the point about ERPS.

15. Fill in any missing doi in your bibliography.

Author Response

Please see the attached document below. 

Round 2

Reviewer 2 Report

Comments and Suggestions for Authors

Thank you for the revised manuscript and for addressing the comments so thoroughly. I appreciate the thoughtful revisions and clarifications provided, which have indeed enhanced the manuscript significantly.

I am happy to confirm that I accept the revised article.

Reviewer 3 Report

Comments and Suggestions for Authors

I would like to report that all my suggestions have been considered and the text has been upgraded. It is important that what I have noted has been addressed.